# CryoEM structure of adenovirus type 3 fibre with desmoglein 2 shows an unusual mode of receptor engagement

Emilie Vassal-Stermann[1], Gregory Effantin[1], Chloe Zubieta[2], Wim Burmeister [1], Frédéric Iseni[3], Hongjie Wang[4], André Lieber[4], Guy Schoehn [1] & Pascal Fender [1]

Attachment of human adenovirus (HAd) to the host cell is a critical step of infection. Initial attachment occurs via the adenoviral fibre knob protein and a cellular receptor. Here we report the cryo-electron microscopy (cryo-EM) structure of a <100 kDa non-symmetrical complex comprising the trimeric HAd type 3 fibre knob (HAd3K) and human desmoglein 2 (DSG2). The structure reveals a unique stoichiometry of 1:1 and 2:1 (DSG2: knob trimer) not previously observed for other HAd-receptor complexes. We demonstrate that mutating Asp261 in the fibre knob is sufficient to totally abolish receptor binding. These data shed new light on adenovirus infection strategies and provide insights for adenoviral vector development and structure-based design.

[1] Institut de Biologie Structurale (IBS), Université Grenoble Alpes, CNRS, CEA, 71 Avenue des Martyrs, 38042 Grenoble, France. [2] Laboratoire de Physiologie Cellulaire et Végétale, Biosciences and Biotechnology Institute of Grenoble, UMR5168, CNRS/CEA/INRA/UGA, 17 Rue des Martyrs, 38054 Grenoble, France. [3] Unité de Virologie, Institut de Recherche Biomédicale des Armées, BP 7391223 Brétigny-sur-Orge Cedex, France. [4] Department of Medicine, Division of Medical Genetics, University of Washington, Box 357720Seattle, WA 98195, USA. These authors contributed equally: Emilie Vassal-Stermann, Gregory Effantin. Correspondence and requests for materials should be addressed to A.L. (email: lieber00@u.washington.edu) or to P.F. (email: pascal.fender@ibs.fr)

Human adenoviruses are common pathogens associated with respiratory, gastrointestinal and ocular infections. HAd can be divided into seven species (A–G) and comprise over 50 disease-causing serotypes. The B species serotype, HAd3, is widespread in Europe, Asia and North America. Recent studies from the United States and Europe show that HAd3 infections occur more often in adolescents and adults, while studies from Asia indicate that HAd3 is prevalent in young children, often causing severe respiratory symptoms[1–4]. Beside the pathogenicity of the virus, adenoviral vectors based on HAd3 are gaining interest as therapeutic agents in cancer virotherapy[5–7]. In order to address both the development of vaccines against HAd3 and the optimization of HAd3 vectors for therapeutic applications, a detailed understanding of HAd- host cell interaction is required.

The first step of adenoviral replication involves the interaction of the fibre protein and more precisely the globular trimeric knob with an attachment receptor from the host cell. Coxsackie and Adenovirus Receptor (CAR) and CD46 have been previously reported to be targeted by many adenovirus serotypes[8–11]. Desmoglein 2 (DSG2), a newly identified adenovirus receptor has been reported to be used by some species B human adenoviruses including HAd3, HAd7, HAd11 and HAd14 for cell infection[12]. While extensive structural studies of adenovirus fibre knob interaction to CAR or CD46 have been reported[13–15], little is known for HAd interaction with the desmosomal cadherin DSG2.

In cells, the three-dimensional organization of native desmosomes have been visualized by cryo-electron tomography studies of vitreous sections[16]. 2D projection images of the extracellular core domain (ECD) revealed a general phenotype with extracellular domains of desmosomal cadherins as electron-dense protrusions with a pronounced periodicity. This study highlighted quasi-periodically arrangement and specific organization of cadherins, alternating *trans* and *cis* interactions. Ab initio modelling into the desmosome tomography maps show parallel rows of desmoglein or desmocollin forming *trans* interactions across the midline. In desmosomes, the type-1 transmembrane protein DSG2 makes heterophilic interactions with desmocollin 2 (DSC2)[17,18]. The structure of the extracellular region of DSG2 containing four cadherin domains, EC1 to EC4, has recently been solved by crystallography[19]. The intermediate portion of the DSG2 ectodomain, consisting of EC2 and EC3, has been described as important for recognition by the HAd3 knob (HAd3K), the trimeric globular distal part of the fibre protein. We recently demonstrated that HAd3K binds to DSG2 by a non-classical mechanism involving mainly one receptor bound per trimeric fibre knob. A second minor complex harboring two receptors per trimeric fibre knob was also found[20]. However, no atomic resolution data were available describing this interaction.

Here, we report the structure of HAd3 fibre knob (HAd3K) in complex with one and two molecules of the DSG2 receptor. The structure was solved using cryo-EM with a phase plate to image the small (<100 kDa) non-symmetrical complexes. Based on these data, we identified the residues critical for HAd-receptor interactions and the structural rearrangements due to HAd3 binding to the receptor.

## Results

**Two non-symmetrical complexes can be observed.** Different population of particles with either one or two EC2-EC3 modules (HAd3K/EC2-EC3 and HAd3K/(EC2-EC3)₂) were identifiable (Fig. 1a–c, Supplementary Fig. 1a–c) in agreement with the 5.40S and 7.34S species previously reported[20]. Their 3D structures were solved to an overall resolution of 3.5 and 3.8 Å, respectively (Supplementary Table 1). The resolution of both 3D maps is relatively uniform, the HAd3K and the EC2-EC3 core regions being slightly better defined than the receptor's distal parts (Supplementary Fig. 1d, e). The atomic coordinates of both HAd3K (PDB: 1H7Z) and EC2-EC3 (PDB: 5ERD) were used to fit the cryo-EM maps and refined (Fig. 1a–c, Supplementary Fig. 1f–k). In the 7.34S complex, one of the two EC2-EC3 modules is bound similarly to the corresponding one of the 5.40S complex. However, the relative positions of the modules differ slightly in the two cryo-EM maps (rmsd of 2.585 Å between the two EC2-EC3 modules) likely due to the binding of the second module in the 7.34S species (Supplementary Table 2). The structural data confirm the biochemical data since clashes occur with three modules excluding the more commonly observed three-receptor per knob binding mode observed for other HAd-receptor complexes (Fig. 1d, e). The 1:1 (receptor: knob) stoichiometry is statistically the preferred binding strategy from the cryo-EM analysis (Supplementary Table 1) and is in agreement with our previous biochemical study[20].

**DSG2 binds the top of HAd3K.** The interaction with DSG2 ectodomains (EC2 and EC3) takes place at the top of the trimeric HAd3K close to the three-fold axis (Fig. 1a–c). The EC2-EC3 domains span two HAd3K monomers, with the third monomer having no contacts with the receptor in the 1:1 complex. Each EC domain interacts with one monomer and targets primarily loop regions of HAd3K. In monomer 'A', the EC2 domain interacts with a surface exposed loop between the A and B strands and the C–D loop and N-terminal portion of the D strand. EC3 interacts with a loop between the G–H strands of monomer 'B'. In the 2:1 complex (DSG2: trimeric knob), the interactions of the 1:1 complex are preserved, with the second DSG domain EC3 interacting with the G–H loop of monomer 'A' and the EC2 domain interacting with the A–B and C–D loop and N-terminal portion of the D strand of monomer 'C'.

Whether one or two receptors are bound, both EC2 and EC3 domains are involved in the interaction (Fig. 2a, b, Supplementary Fig. 2a, b). This observation is in agreement with the biochemical data reporting that EC3 is critical for HAd3K binding while EC2 stabilizes this interaction[20]. Several residues distributed along the EC2 primary sequence were found to interact with the HAd3K monomer, while only one region in EC3 (covering residues 311 to 321) is involved in the interaction with the groove of a second monomer (Fig. 2c). Thus, each DSG2 cadherin domain interact with only a single monomer (Fig. 2a, b) as previously reported for CD46 (SCR1 and SCR2 domains) interaction with HAd11K (Fig. 1d).

This binding mode is in stark contrast with both CD46/HAd11K and CAR/HAd12K interactions, which involve the intermediate or lower part of the knob, respectively[13,14] (Fig. 1d, e). The location of only one (HAd3K/EC2-EC3) or two (HAd3K/(EC2-EC3)₂) receptor module close to the three-fold axis is unusual as compared to CD46 and CAR for which three receptors were found at the periphery of the fibre knob. The broken symmetry of the HAd3K-DSG structures due to the use of different interaction surfaces in each of the HAd3K monomers does not occur in other HAd-receptor complexes in which the three-fold symmetry of the complex is preserved upon receptor binding.

**Identification and mutagenesis of the critical residues.** The main interacting residues are highlighted in the primary sequence of both EC2-EC3 and HAd3K (Fig. 2c, d). Overall, the same HAd3K residues are used for these two forms. The main difference relies on three extra glutamates (E146-E268-E299) from HAd3K, which might be involved in the second module interaction while they do not participate to the first EC2-EC3 module

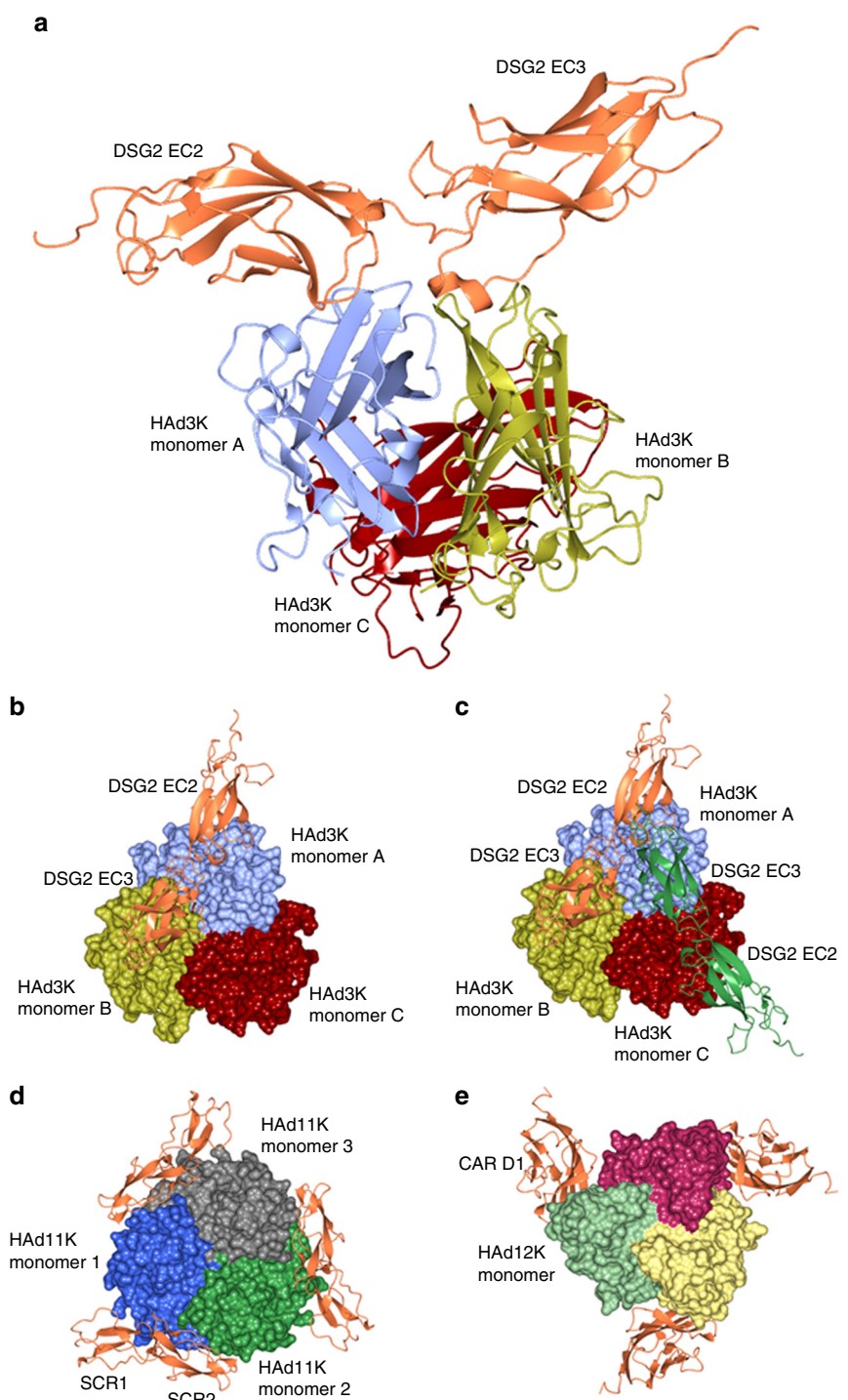

**Fig. 1** Atomic reconstruction of HAd3K/EC2-EC3 complex, comparison to CAR and CD46. **a** Side view of HAd3K/EC2-EC3 complex with the three HAd3K monomers colored in blue, gold and red and EC2-EC3 colored in orange. **b** Top view of the same complex. **c** Top view of HAd3K/(EC2-EC3)$_2$ with the second module in green. **d** Top view of HAd11 Knob structure with three CD46 SCR1&2 molecules in orange (PDB: 2O39). **e** Top view of HAd12 Knob with CAR D1 domain in orange (PDB: 1KAC)

binding (Supplementary Table 2). Previous studies using random mutations in the HAd3K reported that single mutations N186D, V189G, S190P or L296R reduced DSG2 binding by more than 80% and that D261N or F265L totally abolished the binding[21]. Remarkably, all these residues were found to be involved in the interaction with DSG2 in the structures presented here (Fig. 2d, asterisks). N186, V189 and S190 are at the interface between HAd3K and the EC2 module while L296, D261 or F265 are part

of the HAd3K/EC3 module interface. While the modest resolution (3.5 and 3.8 Å) of the cryo-EM reconstructions makes unambiguous characterization of the interactions between HAd3K and DSG2 difficult, several putative hydrogen bonds and salt bridges were identified with PDBePISA (Supplementary Table 2). Examples include S190 (HAd3K) with S175 (DSG2 EC2) and D261 (HAd3K) with R316 (DSG2 EC3), which would stabilize the loop conformation of HAd3K and may be required

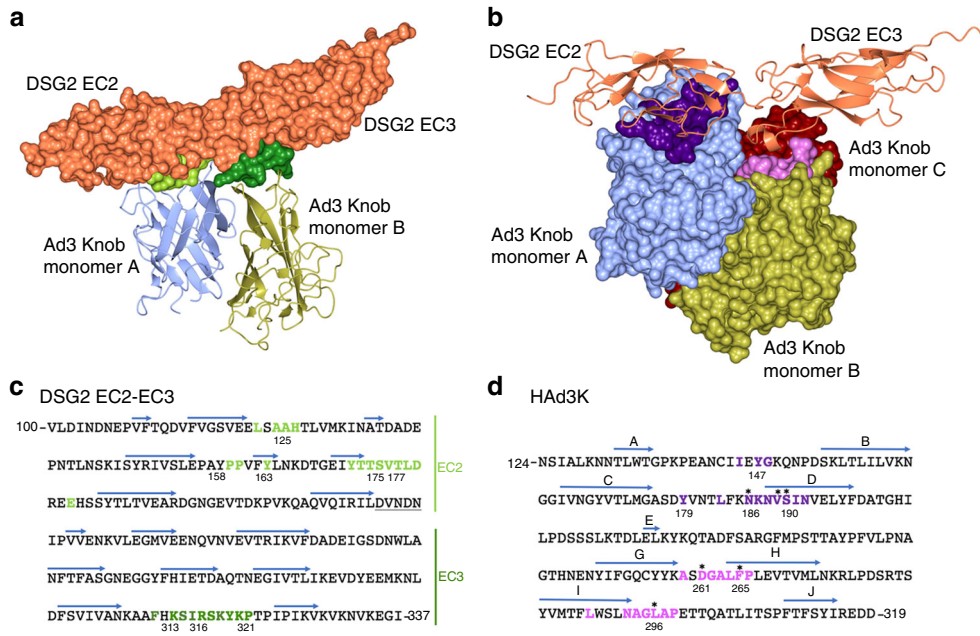

**Fig. 2** Determination of residues from HAd3K and EC2-EC3 involved in the interaction. **a** Surface of EC2-EC3 interacting with HAd3K. Aminoacids making contact with HAd3K are colored in light or dark green according to their interaction with monomer A or B, respectively. For a better view only two monomers of HAd3K are represented in cartoon. **b** Surface of HAd3K involved in receptor binding. HAd3K aminoacids involved in binding are highlighted in dark purple (EC2 contacts) and light purple (EC3 contacts). **c** EC2-EC3 primary sequence with aminoacids making contact with the knob are colored using same color code than in **a**. **d** Primary sequence of HAd3K with β-strand depicted with arrows. Contact-forming residues of HAd3K are depicted with same colors as in **b**. Asterisks (*) denote residues for which single mutation reduce DSG2 binding by more than 80% as decribed in Wang et al.[21]

for DSG2 binding (Supplementary Fig. 2c, d). In order to determine the importance of the D261 (HAd3K) with R316 (DSG2 EC3) interaction, a D261A mutant was produced and incubated with an excess of EC2-EC3 protein. Remarkably, the receptor binding is totally abolished when the D261A mutant versus the wild type protein is used (Fig. 3a, b). This result further supports our previous biochemical data indicating that EC3 binding is critical for complex formation[20].

**Fibre knob binding induces a twist in DSG2 domains**. A feature of the HAdK3/DSG2 interaction is a conformational change between the unbound and bound receptor. When superimposing the cryo-EM (from HAd3K/EC2-EC3 (5.40S)) and structure of the DSG receptor (5ERD) by their EC3 domains (rmsd = 1.007 Å), the EC2 domain rotates upon interaction with HAd3K by ~10° (Fig. 4). This suggests that the relative orientation of EC2 and EC3 is altered upon binding to HAd3K. Since EC3 binding is critical for the interaction as mentioned above and contributes the majority of the binding energy, it is likely this domain interacts first with HAd3K. The observed twist would be induced in a second step for a better fitting of EC2 with the second Ad3K monomer. A similar observation has been made for the binding of CD46 to HAd11K where an even more dramatic change of curvature (~60°) between CD46 SCR1 and SCR2 was observed upon binding. The HAd3K structure, in contrast does not exhibit dramatic rearrangement upon EC2-EC3 binding.

## Discussion

In this work, we used cryo-EM to determine the structure of human Adenovirus 3 fibre knob with the DSG2 receptor EC2 and EC3 domains. Numerous attempts at obtaining well-diffracting crystals of the complex were unsuccessful, likely due to the presence of complexes with different stoichiometry. Despite the low molecular weight (96 kDa) of the 1:1 (HAd3K: EC2-EC3)

complex[20], cryo-EM with a phase plate was used in structure determination resulting in 3.5 Å data for the 1:1 complex and 3.8 Å data for the 1:2 complex (Fig. 1a–c; Supplementary Table 1). The structures presented here represent some of the smallest non-symmetric complexes solved to date using cryo-EM and demonstrate the utility of this technique even on complexes less than 100 kDa. While high-resolution crystallographic studies of proteins and complexes can be considered the gold standard, cryo-EM is fast becoming a viable option for even relatively small macromolecules and complexes.

The structure of this complex clearly differs from other adenovirus fibre knob/receptor complexes characterised to date, differing in both stoichiometry and binding surfaces. When looking from the top along the three-fold axis, the EC2-EC3 domains occupy positions closer to the central three-fold axis of the knob as compared to HAd11K with CD46 or HAd12K with CAR for which the receptors are located at the periphery of the knob domain (Fig. 1b–e)[13,14]. In addition, the non-classical stoichiometry of the HAd3K-DSG complex (1:1 and 1:2) results in non-symmetrical binding of the receptor to the trimeric knob. Indeed, both CAR (D1 domain) and CD46 (SCR1-SCR2 domains) were found with three copies per trimeric knob, preserving the three-fold symmetry of the complex (Fig. 1d, e). Whether or not both these stoichiometries are biologically relevant or important for viral infection is debateable.

The three-dimensional organization of native desmosomal cadherins has been visualized by cryo-electron tomography, revealing a quasi-periodically arrangement with rows of desmoglein and desmocolin forming *trans* interactions across the midline[16]. Modeling of HAd3K binding to one DSG2 molecule in such an organized structure precludes the binding of a second DSG2 molecule coming either in *cis* or in *trans* without significant rearrangement of the desmosomal cadherins. However, a likely consequence of this 1:1 stoichiometric binding might be that a single fiber would be unable to induce signal transduction.

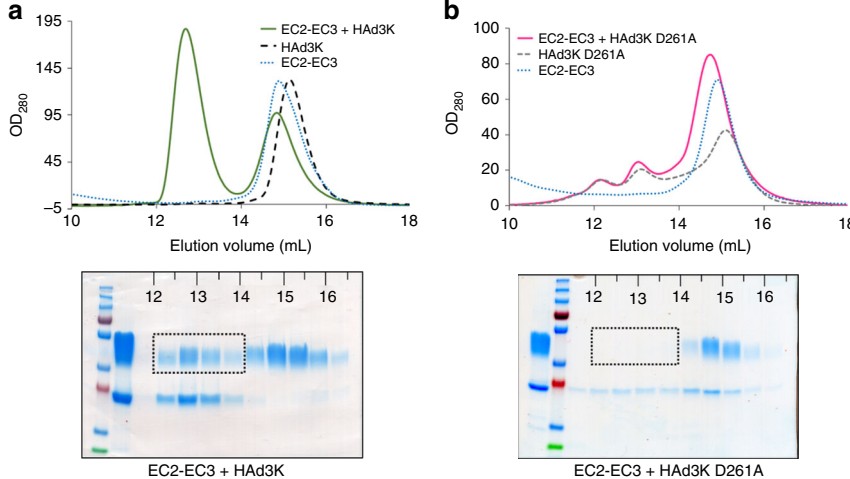

**Fig. 3** A single D261A mutation in HAd3K is sufficient to abolish the receptor binding. **a** HAd3K incubation with an excess of EC2-EC3 produced in mammalian cell resulted in a complex eluted between 12 and 14 mL as shown by SEC and confirmed by SDS-PAGE. **b** Incubation of EC2-EC3 with the D261A mutated HAd3K resulted in a pic eluted latter (14–16 mL) corresponding to the two free components as shown by SDS-PAGE. Dotted rectangles on SDS-PAGE indicates the expected position of EC2-EC3 when in complex with the knob

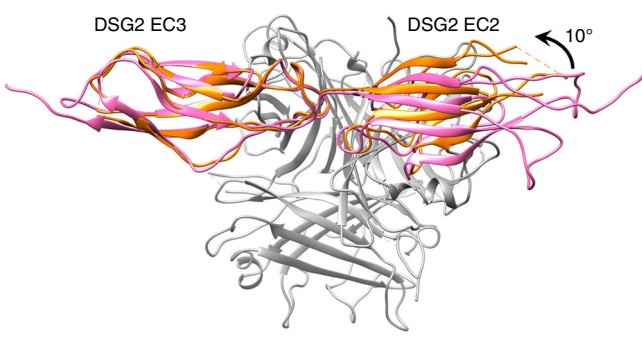

**Fig. 4** Receptor cadherin domains twisting upon HAd3K binding. Superimposition of unliganded (in pink) and bound (in orange) EC2-EC3 to HAd3K (in grey) shows a 10° twist between EC2 and EC3

Interestingly, DSG2-interacting adenoviruses express during their replication cycle a large excess of penton-dodecahedrons (Pt-Dd)[22], small viral-like particles in which twelve fibres are exposed. It has also been reported that HAd3K is unable to induce cell remodeling whereas Pt-Dd (i.e. with 12 fibre knobs per particle) effectively triggered opening of intercellular junctions to facilitate viral spread[23,24]. This suggests an attractive model in which Pt-Dd has evolved to effectively open intracellular junctions by binding several DSG2 receptors at once in a manner compatible with the native desmosomal cadherin arrangement. It is likely that at the cellular level, each HAd3K trimer knob interacts with a single receptor due to (i) the abundance of this form in our in vitro sample (Supplementary Table 1), (ii) the highly-structured organization of desmosomes in which DSG2 forms well-organised parallel zippers, which preclude the binding of a second DSG2 molecule with a compatible angle and (iii) signal transduction has never been observed with single knobs contrary to Pt-Dd, suggesting that multivalent interaction cannot be done by a single HAd3K.

While the structure of 1:1 (fibre knob:receptor) has never been reported for other protein receptors prior to this study, HAd37K binding to the GD1a glycan exhibits a 1:1 binding mode. The hexasaccharide glycan molecule carries two sialic acids that occupy binding sites on two of the three monomers resulting in a 1:1 stoichiometry[25]. In the present structure, however, the interaction surfaces are different and the 1:1 and 1:2 binding

modes are largely dictated by steric hindrance occurring close to the three-fold axis of the HAd3K, likely resulting in less favorable binding of two EC2-EC3 modules binding and extensive clashes if a third EC2-EC3 module is present. While the stoichiometry and non-symmetrical binding of the receptor is unsual for the DSG structure, other HAdK- protein receptor complexes exhibit similarities in binding multiple domains or motifs. In the 1:1 HAd3K-DSG structure only a single monomer of HAd3K binds either the EC2 or EC3 domains (Fig. 2a, b). This type of binding is similar to what has been described for CD46 SCR1 and SCR2 modules interacting with HAd11K in which SCR1 and SCR2 occupy distinct binding sites on each monomer[14].

In both the 1:1 and 1:2 structures, the critical residues from HAd3K involved in the interaction with EC2-EC3 are mostly conserved (Supplementary Table 2). The residues are located primarily in surface exposed loops between beta strands and form putative salt bridges and/or hydrogen bonding interactions with the receptor. The identification of the HAd3K residues involved in the interaction with DSG2 resulted in a striking finding that the majority of them had been identified in a HAd3K random mutagenesis study we had previously performed[21]. The relatively modest 3.5–3.8 Å resolution made unambiguous identification of interacting amino acids challenging; however, the complementarity of our structural and biochemical data provides strong evidence that we have determined the interacting regions in the complex.

Both our mutation and structural studies highlighted that D261 in HAd3K is critical for the interaction with EC3 and thus with DSG2. To confirm this, a D261A mutation was introduced and the HAd3K^D261A protein was incubated with an excess of EC2-EC3. Size Exclusion Chromatography (SEC) and SDS-PAGE analysis (Fig. 3a, b) showed that no stable complex between HAd3K^D261A and EC2-EC3 was formed, only the free components were seen for the D261A mutation. This result combined to our previous reports showing that EC3 containing module (i.e. EC3-EC4) can bind HAd3K contrary to the EC2 containing module (i.e. EC1-EC2) reinforces the critical role played by the EC3 domain in HAd3K binding. EC3 is likely to be targeted first by HAd3K with D261 playing a crucial role in the interaction and a rearrangement in DSG2 would then occur to enable the interaction of EC2 with the adjacent HAd3K monomer thus explaining the 10° twist observed between EC2 and EC3 once bound to HAd3K (Fig. 4). This is likely as the presence of EC2 increases the binding affinity of the complex. Similar conclusions

have already been reported for HAd11 binding to CD46. Indeed, HAd11K engagement of CD46 triggers a conformational change between the SCR1 and SCR2 domains resulting in a nearly linear conformation different from the kinked conformation of the unliganded receptor[14]. Moreover, if HAd11K mutation in R279 resulted in a total abolishment of CD46 binding, this residue was not conserved in HAd21K (replaced by a serine) thus requiring a different conformational change in HAd21K and resulting in a lower affinity[15]. Interestingly, position D261 in DSG2-interacting adenoviruses fibre knobs is strictly conserved (HAd3, HAd7, HAd11, HAd14—PDB: 1H7Z, 3EXW, 3EXV, 3F0Y) suggesting that a loss of DSG2 interaction might be obtained by mutating this residue in all DSG2-interacting serotypes. This would provide a simple and efficient way to design adenoviral vectors derived from these serotypes that detarget DSG2 if retargeting to another receptor is needed.

The data presented here reveal a non-classical mode of inter-action for DSG2-interacting adenoviruses and provides new clues for the rational design of DSG2-interacting adenoviruses, subviral particles and fibre knobs. Such vectors are currently undergoing a rapid development in therapy with HAd3 and HAd11 currently in clinical trials[5–7]. Beside the use of oncolytic adenoviruses, HAd3 fibre-containing molecules, such as penton-dodecahedron (symmetric particle harboring 12 fibres) or junction-openers (JOs), have been reported to act as enhancers of approved treatments in cancer therapies[12,26–29]. The data presented here provide important information for structure-based design of drugs targeting the HAd-DSG2 interaction for anti-virals and will allow the optimization of DSG2 adenoviral vectors for therapeutic applications.

## Methods

**Complex expression and purification**. HAd3K (fibre residues 124–319) and EC2-EC3 domains (DSG2 residues 100–337) sequences were cloned into a pETDuet vector (Novagen) and co-expressed in *E. coli* Rosetta™(DE3)pLysS cells (Novagen). HAd3K was inserted in MCS-1, yielded a recombinant protein with a hexa-histidine tag at the N-terminus whereas DSG2 EC2-EC3 domains was introduced in MCS-II with no fused affinity tag. All the primer sequences can be found in Supplementary Table 3. Cells pellets containing the two recombinant proteins were resuspended in buffer containing 500 mM NaCl, 3 mM CaCl2, 10 mM Imidazole, 20 mM Tris-Cl pH8.0 and cOmplete™ EDTA-free Protease Inhibitor Cocktail. Cells were lysed by sonication. Clarified supernatant obtained after centrifugation was loaded onto a 1-mL His GraviTrap prepacked column (GE Healthcare). HAd3K/EC2-EC3 complex bound to the Ni–NTA column were eluted with buffer con-taining 500 mM sodium chloride, 3 mM calcium chloride, 20 mM Tris-Cl pH 8.0 and 200 mM imidazole. The complex was further purified on a Superdex™ 200 Increase 10/300 GL column (GE Healthcare). Purified complex was concentrated to 7 mg/mL in buffer containing 150 mM NaCl, 3 mM CaCl₂, 10 mM Tris-Cl pH 8.0.

For the complex formation study with HAd3K and D261A-mutated HAd3K, EC2-EC3 was produced in HEK293 cell as previously reported[20]. The complex was formed by incubating 210 µg of either HAd3K or D261A HAd3K with 260 µg of EC2-EC3 in a final volume of 300 µL of 150 mM sodium chloride, 3 mM calcium chloride, 20 mM Tris-Cl pH 8.0. Samples were analyzed on Superdex™ 200 Increase 10/300 GL column (GE Healthcare).

**Cryo-electron microscopy**. Three microlitre sample were applied to 1.2/1.3 Quantifoil holey carbon grid (Quantifoil Micro Tools GmbH, Germany) and the grids were plunged frozen in liquid ethane with a Vitrobot Mark IV (Thermo Fisher Scientific) (2–3 s blot time, blot force 0). The sample was observed with a Titan Krios (Thermo Fischer Scientific) at 300 kV with an energy filter (Quantum LS, Gatan Inc, USA) (slit width of 20 eV) and a Volta phase plate. Images were recorded automatically on a K2 summit direct detector (Gatan Inc., USA) in super resolution counting mode with EPU (Thermo Fisher Scientific). Movies were recorded for a total exposure of 7 s and 175 ms per frame resulting in 40 frame's movies with a total dose of $\sim$35 e$^-$/Å$^2$ (dose of $\sim$1.5 e$^-$/pixel/s) The magnification was 130,000 (0.53 Å/pixel at the camera level). The defocus of the images varies between $-0.5$ and $-1.0$ µm. The total dataset was acquired during two distinct sessions on two different grids prepared from the same protein purification. The phase plate position was changed automatically every 100 images, which corresponds to an accumulated dose of less than 100 nC for each phase plate position.

**3D reconstruction**. The movies were first drift-corrected with motioncor2 without using the first two frames[30]. The remaining image processing was done in RELION 2.1[31]. CTF estimation was done with GCTF[32]. An initial set of particles (box size of 100 pixels, sampling of 2.12 Å/pixel) was obtained by autopicking with a gaussian blob. After 2D classification (run1), the best looking 2D class averages were used as references for a second autopicking. Following two more 2D classifications (run2 and 3), 2D class averages displaying either one or two DSG2 molecules were clearly identifiable. Particles with one DSG2 module were selected for a first 3D refinement using, as an initial model, a low-pass filtered (at 40 Å) density map calculated from the crystal structure of the HAd3K (PDB 1H7Z)[33]. This first 3D map was then used as a reference for 3D classification in three classes. A more homogeneous popu-lation of particles with one DSG2 was isolated and an improved 3D map was then calculated.

In order to better separate particles with one or two DSG2, a 3D classification (in five classes) of the particles belonging to all the good looking 2D class averages from run3 (see above) was done using the 3D map obtained for one DSG2 as a reference. This allowed the isolation of particles with one or two DSG2, which were then refine to obtain two distinct 3D reconstructions of HAd3K with either one or two DSG2 modules.

All the above steps were done with the data acquired during the first session. Essentially the same steps were followed for the analysis of the data acquired during the second microscope session.

Finally, the two data sets were combined and particles were re-extracted with a box size of 200 pixels (final sampling of 1.06 Å/pixel). Local defocus was estimated for each particle with GCTF. Following a last 3D classification in three classes (for both particles with one or two DSG2), the two final 3D maps for HAd3K in complex with one or two DSG2 modules were calculated from 139958 and 78925 particles, respectively. The resolutions of these two 3D maps determined by Fourier Shell Correlation (at FSC = 0.143) were 3.5 and 3.8 Å, respectively, after post processing in RELION.

**Model refinement**. The crystal structures of the HAd3K (PDB 1H7Z) and of domains EC2-EC3 of DSG2[19] (PDB 5ERD) were rigid-body fitted inside the cryo-EM density maps in CHIMERA[34]. The atomic coordinates were then refined with ROSETTA[35] and PHENIX[36]. The refined atomic models were visually checked and adjusted (if necessary) in COOT[37]. The final models for HAd3K with either one or two DSG2 were validated with MOLPROBITY[38]. Amino acids for which there were no clear densities were not included in the model refinements (~9 and ~11% of the total residues for HAd3K with one or two DSG2. respectively).

Analysis of the interaction between the DSG2 modules and HAd3K were done in CHIMERA[34] and PISA[39] ('Protein interfaces, surfaces and assemblies' service PISA at the European Bioinformatics Institute. (http://www.ebi.ac.uk/pdbe/prot_int/pistart.html)). Any putative interaction identified with PISA for which the distance between atoms was more than 3.5 Å for hydrogen bond and 4 Å for salt bridge was discarded. The figures were prepared with CHIMERA. The data collection and model statistics are summarized in Supplementary Table 1.

**Reporting summary**. Further information on experimental design is available in the Nature Research Reporting Summary linked to this article.

## Data availability

The authors declare that the data supporting the findings are available from the corresponding authors upon reasonable request. Coordinates for the two complexes have the PDB accession codes 6QNT (one EC2-EC3 module) and 6QNU (two EC2-EC3 modules). The two cryo-EM maps have the EMDB accession codes EMD-4608 and EMD-4609, respectively.

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

## Acknowledgements

We are grateful to the ANR for its support to the 'Ad-Cadh' project ANR-18-CE11-0001. We acknowledge access to the European Synchrotron Radiation Facility (ESRF), and thank CM01 scientists for their help with data collection. This work used the platforms of the Grenoble Instruct-ERIC Center (ISBG: UMS 3518 CNRS-CEA-UGA-EMBL) with support from FRISBI (ANR-10 INBS-05-02) and GRAL (ANR-10-LABX-49-01) within the Grenoble Partnership for Structural Biology (PSB). We thank Daphna Fenel and Emmanuelle Neumann for preliminary negative staining screening. The electron microscope facility is supported by the Rhône-Alpes Region, the Fondation Recherche Medicale (FRM), the Fonds Européen de Développement Régional (FEDER) and the GIS-Infrastrutures en Biologie Sante et Agronomie (IBISA).

## Author contributions

A.L. and P.F. supervised the project. E.V.S., G.E., F.I. and G.S. designed the experiments. E.V.S., G.E., C.Z., H.W. and W.B. performed the biochemical and structural experiments. E.V.S., G.E., C.Z., G.S., A.L. and P.F. wrote the manuscript.

## Additional information

**Competing interests:** The authors declare no competing interests.

