## [Peer Review File · Nature Communications]

Reviewers' comments:

Reviewer #1 (Remarks to the Author):

Understanding the interaction between a virus and its primary receptor is critical for understanding the fundamental mechanisms of viral entry into a host cell. Cryo-EM structures are highly informative to identify critical interactions and potential structural changes that can induce cellular signaling. This manuscript describes the structural interaction between the adenovirus type 3 fiber knob with its receptor, desmoglein-2. The major findings that the interaction is preferentially 1:1, occurs at the top of the trimeric HAd3K, and induces a twist into the extracellular domain of desmoglein-2 by binding the second and third extracellular domain are important. However there are several concerns with this manuscript.

1. The explanation for resolving the 1:1 binding preference, which is in agreement with biochemical studies but disagreement with the requirement for multimerization for initiation of intracellular signaling requires clarity.
2. The data for hAd3K interaction with one DSG2 module requires explanation. Ramachandran outliers are considerably worse in comparison to other EM scores, highlighted particularly in chain D (quality of 95% compared to 97-98% of other chains). Chain D contains four Ramachandran outliers (8th percentile with respect to all PDB EM entries), calling into question the validity of the structure reconstruction. However, the standard geometry of the structure is fine, noted by the minimal clashes and lack of other issues in the bond length/angle parameters, atom chirality or planarity.
3. The data for hAd3K interaction with two DSG2 modules requires explanation. While the EM structure has acceptable standard geometry and clash statistics, this structure is also concerning in regards to Ramachandran outliers. In addition to EC2-EC3 outliers, outliers now also exist within the fiber chain A. The Ramachandran percentiles with respect to all PDB EM entries are very low, ranging from 7-16th, giving a total structure percentile of 16 in regards to Ramachandran outliers. With the exception of Ramachandran outliers, structures, as well as experimental methods for structure determination, are adequate.
4. The manuscript needs significant editing.

Katherine Excoffon

Reviewer #2 (Remarks to the Author):

In this study, Vassal-Stermann et al. examine the structure of a protein complex formed by the trimeric human adenovirus of type 3 fibre knob (HAd3K) and human desmoglein 2 (DSG2) by using phase plate cryo-EM. With PP cryo-EM the authors is able to show the structure of HAd3K bound with one or two receptors. The structure reveals the unique features of the receptor interactions, shedding light to new adenovirus infection strategies. However, the manuscript lacks depth in describing only the complex structures. It is lacking in a biological "story" revealed by the structures instead of just a structure description.

Especially, the structural description of the complex between HAd3K and the two receptors is not clear in the manuscript. In Fig. 1b, the authors should also show the top view model of the two EC2-EC3 modues, and explain why two modules can bind to HAd3K, not three. Furthremore, if the interaction of the second receptor for HAd3K is defferent from the first one, the authors should described it. Then, it is necessary to describe the biological meaning of the difference between one module and two modules.

Minor comments:

- 1) Line 53 5,40S -> 5.40S
- 2) Line 54 7,34S -> 7.34S
- 3) Line 76 5,40S -> 5.40S
- 4) Line 77 7,34S -> 7.34S
- 5) In Fig. S2c and d, it is difficult to distinguish between HAd3K density and EC2-EC3 module density. The maps must be displayed in a different color between HAd3K and EC2-EC3 module.
- 6) In method (Supplemental information), the exposure dose is described as ~ 6 e⁰/pixel/s, but it is ~ 1.5 e⁻/pixel/s in the table. Please check.
- 7) 3D reconstruction (Supplemental information)
Line 1 the first two frame -> the first two frames

Reviewer #3 (Remarks to the Author):

The short paper submitted by Vallas-Stermann et al. provides new structural insights on the interaction between the hAdV-B3 fiber knob domain and its recently elucidated receptor, desmoglein 2. Species B1 adenoviruses are increasingly important as therapeutic agents, in particular in the cancer arena, where they are under development as oncolytic agents – e.g. Ad5/3 chimeric vectors and Enadenotucirev, a HAdV-B3/11 hybrid vector. Therefore, new insights into the virus: host receptor interactions are important and highly relevant to the field. To date, a structure for the complex between species B hAdVs and DSG2 has been lacking, and therefore the present study is novel and of importance to the field.

The current, extremely brief article appears to have been transferred to Nature Comms from another nature journal and has not undergone any reformatting at this stage for the journal. This makes it somewhat difficult to review properly at this stage, and the first stage of any resubmission must involve significant reformatting to better align with the readership of Nature Communications. The authors will have a significantly extended word limit and should therefore use this to better outline the field, adenovirus structure and function, and place their research within the wider context. In addition to this very broad comment, which should be addressed, I have several additional comments which I think would help to strengthen a resubmitted manuscript.

- Given that the resolution achieved here (3.2 – 3.8 angstrom) is relatively low resolution. Whilst I understand that this is likely a feature of the technique used (i.e. cryoelectron microscopy). It would be useful to know why the authors chose this technique for their analysis when the “gold standard” technique used to determine higher resolution structures is crystallographic analysis. If the authors chose not to use this technique, then their complex will remain low resolution. It is possible (perhaps probable) that given the experience of the investigators in crystallography, that this simply wasn’t possible in this case. Should this be the case then the authors should at a minimum address this limitation in the revised manuscript.
- The authors provide interesting analysis of the hAdV-B3 complex with DSG2 and demonstrate that the complex can form in a stoichiometry of 1 trimer: 1 DSG2 and also (to a lesser extent) 1 trimer: 2 DSG2 molecules. The data for the 1:2 model should be included in the main manuscript, and it would be interesting to know whether the binding residues are altered in the 1:2 model compared with the 1:1 model, where potential binding residues are highlighted (figure 2e). Please include such an analysis for the 1:2 model.
- I recommend caution around definitive statements on the binding residues, given the low-resolution structures presented. To be definitive, higher resolution complexes would need to be studied, that have been energy minimised (it is not clear from the present manuscript whether energy minimisation analysis has been performed on the structures presented). In addition, inspection of the PDB files presented suggests that some of the some of the interacting residues are more than 3.5 angstroms apart, which is further than would normally be considered possible for such interactions. Again, this could be an artefact of the low-resolution model, but the authors should clarify how they determined these interacting residues and provide a figure for this.
- The “hinging” of DSG2 when in complex with aAdV-B3K (exacerbated further by the binding of

an additional DSG 2 molecule) mirrors similar observations with hAdV-B21 and hAdV-B11 when complexed with CD46 – see <https://www.ncbi.nlm.nih.gov/pubmed/20071571>. It would be informative if the authors were to comment on this and the similarities/differences in any revised version.

- Two very minor points – line 71 – I would suggest that this is “consistent with” rather than “remarkable”, and on line 85, please find a better word than “anyhow” to begin the sentence.

To Reviewer 1 (Katherine Excoffon):

Dear Dr Excoffon, we would like to thank for your positive comments, for recalling the importance of understanding the fundamental mechanism of viral entry and for highlighting the importance of our findings: the unusual 1:1 stoichiometry, the location of the interaction at the top of the fibre knob, the DSG2 mapping and the receptor twist induces upon binding. We have now addressed your concerns in this revised version.

1. *The explanation for resolving the 1:1 binding preference, which is in agreement with biochemical studies but disagreement with the requirement for multimerization for initiation of intracellular signaling requires clarity.*

Author response:

We too found this interesting and have more clearly addressed this issue in the revised manuscript. Our previous work demonstrated that HAd3K (as a single component) was not able to induce cell remodeling while its natural multimerization on penton-dodecahedron (Pt-Dd) (Fender *et al.*, JVI 2012; PMID:22345476) or its artificial dimerization on Junction Openers (JOs) can trigger a cellular response (Wang *et al.*, JVI 2011; PMID: 21525338 – Beyer *et al.*, Cancer Res 2011; PMID: 21990319). This observation indicates that several DSG2 receptors have to be engaged in close proximity via multiple HAd3K molecules. A single HAd3K trimer is able to bind the DSG2 receptor (EC2-EC3 domains), but this binding is insufficient for signaling.

For a better clarity, we have added in the discussion section (**lines 197-213**) a text indicating that the 1:1 scenario is the most likely and noting that multimerization would occur in the biological context of viral infection in which the viral particle or penton-dodecahedron presents multiple fibre knobs.

2. *The data for hAd3K interaction with one DSG2 module requires explanation. Ramachandran outliers are considerably worse in comparison to other EM scores, highlighted particularly in chain D (quality of 95% compared to 97-98% of other chains). Chain D contains four Ramachandran outliers (8th percentile with respect to all PDB EM entries), calling into question the validity of the structure reconstruction. However, the standard geometry of the structure is fine, noted by the minimal clashes and lack of other issues in the bond length/angle parameters, atom chirality or planarity.*

Author response:

We agree with you and we have further refined our previous model to improve the Ramachandran outliers as you can see in the new validation report. Moreover, the validity of the structure is reinforced by our previous paper (Wang *et al.*, JVI 2013, PMID: 23946456) in which some random single mutations in HAd3K resulted in a decrease of DSG2 recognition. All those residues were identified in our Cryo-EM structure, this point is discussed in the ‘results’ and ‘discussion’ section (**lines 135-141 and 231-237**). In addition, to definitely validate our structure we have added an extra experiment using a mutated HAd3K (D261A) incubated with EC2-EC3 expressed in human cell. This experiment confirmed that this single D261A mutation (new Figure 3) is sufficient to abolish the interaction (**lines 146-151 and 238-243**). We do think this result is of high importance for people working on the tropism of adenoviral vectors.

3. *The data for hAd3K interaction with two DSG2 modules requires explanation. While the EM structure has acceptable standard geometry and clash statistics, this structure is also concerning in regards to Ramachandran outliers. The Ramachandran percentiles with respect to all PDB EM entries are very low, ranging from 7-16th, giving a total structure percentile of 16 in regards to Ramachandran outliers. With the exception of Ramachandran outliers, structures, as well as experimental methods for structure determination, are adequate.*

Author response:

We have refined our previous model for the two modules complex in order to improve the Ramachandran outliers as you can see in the new validation report of the two-module complex.

4. *The manuscript needs significant editing*

Author response: As stated in the general comments to the three reviewers, the first version of this paper was transferred swiftly to Nature Communications from a NSMB 'brief communication' format. We agree that editing was needed and we have significantly altered the text to this effect.

We think that this new version takes into account your main concerns and we thank you again for your time and your constructive advice which enabled us to significantly improve our manuscript.

To Reviewer 2:

We would like to thank you for highlighting that our work on a very small complex solved by phase plate cryo-EM reveals the unique features of the receptor interaction shedding light to new adenovirus infection strategies. However, you are concerned by the lack of a biological story and you would like to have a clearer explanation for the two-receptors complex and understand why three EC2-EC3 modules cannot be accommodated.

Author response:

We have now expanded the text in accordance with the Nature Communications format in order to better present our experiments in the context of the biology of the virus and its potential therapeutic applications.

Concerning the biological meaning of the difference between one and two modules, we agree that this point should be more fully addressed in the manuscript. We have taken the opportunity of having an extended word limitation to address this crucial point in greater detail (**lines 197-213**).

In Figure 1d, the author should also show the top view model of the two EC2-EC3 modules and explain why two modules can bind not three.

Author response:

We agree that Figure.1 should show the two complexes from the top view. Figure 1 has been changed accordingly (Figure 1. b-c). We have also discussed why three modules cannot be accommodated on the same HAd3K due to steric clashes (**lines 95-97 and 214-221**).

For your information : models of three EC2-EC3 modules on HAd3K (in grey) on either the 5.40S (left) or 7.34S (right) complexes showed that clashes occurred in both cases as depicted by circles.

Minor comments:

Corrections 1, 2, 3, 4: Done

Correction 5: FigS2.c and S2.d have been redone to better distinguish HAd3K and EC2-EC3

Correction 6: The real value $1,5^e/\text{pixel/s}$ has been corrected in the methods section

Correction 7: Done

We hope that this revised version takes into account your concerns and that you will be satisfied by our modifications. We thank you again for your time and constructive remarks.

To Reviewer 3

We would like to thank you for highlighting the interest of our paper in the context of emerging oncolytic therapies using HAd5/3 or HAd3/11 chimeric adenoviruses as vector and for stating that HAdV/DSG2 structure was lacking and therefore our study is novel and of importance to the field.

Broad comment:

The current, extremely brief article appears to have been transferred to Nature Comms from another nature journal.... The authors will have a significantly extended word limit and should therefore use this to better outline the field, adenovirus structure and function, and place their research within the wider context.

Author response:

This revised manuscript expands on these points as suggested. (Introduction and Conclusion sections)

Specific point 1: Cryo-EM versus gold standard crystallographic analysis.

Author response:

We have struggled a lot with crystallographic analysis for two years. Crystals were obtained in different conditions but diffraction never went higher than 5-10 Å. We tried to improve the crystal quality by various means including additive screening and limited proteolysis without success. We hypothesize that the non-usual 1:1 stoichiometry coupled to the presence of a minor 1:2 (HAd3K : EC2-EC3) complex could be responsible for the poor diffraction of the crystals. We have added a sentence in the discussion section to address this point (**lines 173-175**).

Specific point 2: Data for the 1:2 model should be included in the main manuscript and it would be interesting to know whether the binding residues are altered in the 1:2 model compared with the 1:1 model.

Author response:

These specific points were not discussed in the previous version of our manuscript due to the lack of space. As recommended, we have added the 1:2 model in Figure 1c. Moreover, a new table (Supplementary Table 2) shows that most of the interacting residues are conserved in the two complexes although there are some putative exceptions. For instance, three glutamate residues (E146, E268, E299) from HAd3K are detected in close proximity of the second EC2-EC3 module whereas they are not used by the first EC2-EC3 module. This statement is added in this new version in complement to this supplementary Figure 2 (**lines 132-135**). As the resolution of the structures is modest, we prefer to be conservative in our discussion of interacting residues. Globally, the residues are the same in the 1:1 and 1:2 complex.

Specific point 3: Some of the interacting residues are more than 3.5 Å apart, author should clarify how they determined these residues and provide figure for this.

Author response:

We agree that this point is of importance. We have moderated our conclusions (**lines 141-144**) and in the 'methods' section we explain than any putative interaction for which the distance between atoms was more than 3.5 Å for hydrogen bonds or 4 Å for salt bridge was discarded (**lines 360-362**). A new table (Supplementary Table 2) was added to provide a better explanation on the common and specific interactions taking place in both the 1:1 and 1:2 (HAd3K : EC2-EC3) models.

Specific point 4: The hinging of DSG2 mirrors similar observation with HAd21. It would be informative if the authors were to comment similarities/differences in the revised version.

Author response:

This is a very good suggestion. Indeed, both similarities and differences can be found between CD46 engagement by HAd11 or HAd21 and DSG2 engagement by HAd3. This point is now discussed in this revised version (**lines 249-256**). This comparison gives an added value to our structural description while highlighting the excellent work previously performed on CD46.

Minor points: 'consistent with' 'anyhow' are corrected

We hope to have fully addressed your concerns and we thank you again for your constructive remarks which enabled us to better highlight our findings in a broader context.

REVIEWERS' COMMENTS:

Reviewer #1 (Remarks to the Author):

Strong revisions and conclusions strengthened by additional data and analysis. All concerns addressed

Katherine Excoffon

Reviewer #3 (Remarks to the Author):

The authors have addressed my concerns appropriately, and have added new data concerning the Asp261 mutant data to the manuscript, which adds validity to the structural observations.

I am happy to recommend publication of the revised manuscript.

Alan Parker